# Interferon Beta Activity Is Modulated via Binding of Specific S100 Proteins

**DOI:** 10.3390/ijms21249473

**Published:** 2020-12-13

**Authors:** Alexey S. Kazakov, Alexander D. Sofin, Nadezhda V. Avkhacheva, Alexander I. Denesyuk, Evgenia I. Deryusheva, Victoria A. Rastrygina, Andrey S. Sokolov, Maria E. Permyakova, Ekaterina A. Litus, Vladimir N. Uversky, Eugene A. Permyakov, Sergei E. Permyakov

**Affiliations:** 1Institute for Biological Instrumentation, Pushchino Scientific Center for Biological Research of the Russian Academy of Sciences, Institutskaya str., 7, 142290 Pushchino, Russia; fenixfly@yandex.ru (A.S.K.); AlSofin@mail.ru (A.D.S.); avkhacheva@gmail.com (N.V.A.); adenesyu@abo.fi (A.I.D.); janed1986@ya.ru (E.I.D.); certusfides@gmail.com (V.A.R.); 212sok@gmail.com (A.S.S.); mperm1977@gmail.com (M.E.P.); ealitus@gmail.com (E.A.L.); epermyak@yandex.ru (E.A.P.); 2Structural Bioinformatics Laboratory, Biochemistry, Faculty of Science and Engineering, Åbo Akademi University, 20520 Turku, Finland; 3Department of Molecular Medicine and USF Health Byrd Alzheimer’s Research Institute, Morsani College of Medicine, University of South Florida, Tampa, FL 33612, USA

**Keywords:** cytokine, interferon, S100 protein, protein–protein interaction, cancer

## Abstract

Interferon-β (IFN-β) is a pleiotropic cytokine used for therapy of multiple sclerosis, which is also effective in suppression of viral and bacterial infections and cancer. Recently, we reported a highly specific interaction between IFN-β and S100P lowering IFN-β cytotoxicity to cancer cells (Int J Biol Macromol. 2020; 143: 633–639). S100P is a member of large family of multifunctional Ca^2+^-binding proteins with cytokine-like activities. To probe selectivity of IFN-β—S100 interaction with respect to S100 proteins, we used surface plasmon resonance spectroscopy, chemical crosslinking, and crystal violet assay. Among the thirteen S100 proteins studied S100A1, S100A4, and S100A6 proteins exhibit strictly Ca^2+^-dependent binding to IFN-β with equilibrium dissociation constants, *K_d_*, of 0.04–1.5 µM for their Ca^2+^-bound homodimeric forms. Calcium depletion abolishes the S100—IFN-β interactions. Monomerization of S100A1/A4/A6 decreases *K_d_* values down to 0.11–1.0 nM. Interferon-α is unable of binding to the S100 proteins studied. S100A1/A4 proteins inhibit IFN-β-induced suppression of MCF-7 cells viability. The revealed direct influence of specific S100 proteins on IFN-β activity uncovers a novel regulatory role of particular S100 proteins, and opens up novel approaches to enhancement of therapeutic efficacy of IFN-β.

## 1. Introduction

Interferon beta (IFN-β) is a pleiotropic secreted α-helical cytokine of type I interferon family produced in response to pathogens through stimulation of Toll-like receptors. IFN-β is expressed by a variety of cells, such as plasmacytoid dendritic cells, fibroblasts, monocytes, macrophages, B cells, intestinal epithelial cells, astrocytes, microglia, neurons, etc. [1,2,3] IFN-β acts by binding to cell surface receptors, IFNAR1-IFNAR2 complex or IFNAR1 alone, thereby activating innate and adaptive immune responses by modulating the expression of numerous interferon-stimulated genes via JAK-STAT or alternative signaling pathways [4,5,6]. This gives rise to a broad spectrum of biological effects, including immunomodulatory, anti/pro-inflammatory, antitumor, antiviral and anti/pro-microbial activities [3,7,8,9,10]. IFN-β and its derivatives are an important treatment option for relapsing remitting multiple sclerosis (MS), clinically isolated syndrome and secondary progressive MS, due to their ability to attenuate the relapse rate and slow progression of the disease [11]. Despite promising results of IFN-β use in animal models of rheumatoid arthritis (RA), clinical trials of IFN-β showed no improvements in active RA patients [12]. IFN-β is considered as a driver of dermatomyositis activity [13]. A bioinformatics study demonstrated a major role of IFN-β in progression of systemic lupus erythematosus [14]. IFN-β alone or in combination with ribavirin was shown to be effective for treatment of specific cases of hepatitis C, including certain difficult-to-treat patients [15]. Site-specifically PEGylated IFN-β was proposed for therapy of chronic hepatitis B infection [16]. IFN-β-1b combined with lopinavir-ritonavir and ribavirin is considered as an effective SARS-Co-V-2 treatment for the mild to moderate disease [17]. IFN-β-1a therapy was reported to decrease mortality among patients with Ebola virus by a factor of 1.5–1.9 [18]. In vitro studies suggest intermittent prophylactic use of exogenous IFN-β for prevention of virus-induced exacerbations of asthma and chronic obstructive pulmonary disease [19]. Despite direct antimicrobial activity of IFN-β [20], it favors *Mycobacterium tuberculosis* and *Salmonella enterica* serovar Typhimurium survival in host cells [21,22]. Due to immunomodulatory and antiproliferative activities of IFN-β, its ability to stimulate numerous genes mediating tumor cell apoptosis, to repress cancer stem cells, inhibit angiogenesis and metastasis, it is expected that IFN-β used alone as a monotherapy or in combination with radiation, chemotherapy, genomic methylation, checkpoint inhibitors or monoclonal antibodies (mAbs) may exert antitumor effects [23,24]. For example, in murine tumor models of breast, skin, and lung cancers, conjugation of mAbs with IFN-β drastically increased their antitumor efficiency [25]. Clinical trials confirmed beneficial outcomes of the IFN-β use in combination therapy for patients with metastatic/advanced breast cancer [24]. However, pro-survival protein products of interferon-stimulated genes can lead to cellular resistance and suppression of the antitumor effects [23]. Another factor limiting therapeutic use of IFN-β is an onset of systemic (fever, chills, headache, fatigue, depression, liver dysfunction, etc.) and cutaneous (injection-site inflammation, panniculitis, cutaneous necrosis, ulceration, maculopapular rash, cutaneous vasculitis, psoriasis, sclerosing skin disorders, etc.) side-effects [26,27,28]. Hence, an insight into regulatory mechanisms of IFN-β functioning could give us a useful clue for elaboration of means for compensation of the negative consequences of IFN-β treatment, enhancement of its therapeutic efficiency or novel modalities of IFN-β therapy.

The strongest interaction reported for IFN-β corresponds to its binding to the extracellular domain of IFNAR2 with equilibrium dissociation constant, *K_d_*, below 0.1 nM [29]. A similarly low *K_d_* of 0.3 nM was recently reported for IFN-β binding to monomeric S100P protein [30], a cancer-related member of the S100 protein family [31]. Since S100P binding lowers IFN-β cytotoxicity to MCF-7 breast cancer cells [30], S100P inhibition could promote enhancement of anticancer activity of IFN-β. Meanwhile, S100 protein family contains 25 cytosolic/nuclear/extracellular structurally homologous, but functionally diverse, Ca^2+^-binding proteins of the EF-hand superfamily, some of which also bind Zn^2+^/Cu^2+^/Fe^2+^ [32,33,34]. S100 proteins exhibit tissue-specific expression profiles and broad functional repertoire, affected by their metal-binding, posttranslational modifications, oligomerization, and interaction with a wide spectrum of targets. Most human S100 genes are clustered on chromosome 1q21 locus in the epidermal differentiation complex [35], which is the most rapidly evolving gene cluster in humans [36], which is frequently rearranged in cancer [37]. Therefore, many S100 proteins are associated with various types of cancer and other diseases, such as inflammatory, autoimmune, neurological and cardiovascular disorders [33,38,39,40]. Similar to cytokines, some S100 proteins under pathological conditions are translocated/released into the extracellular space, where they act in an autocrine/paracrine manner as damage-associated molecular patterns (“alarmins”) by recognition of specific receptors [30,32]. Considering structural similarity of S100 proteins, one may expect that S100 family members other than S100P could also directly affect functional activities of IFN-β. Here, we probe this hypothesis using a panel of human S100 proteins and MCF-7 cell lines. The revealed capability of specific S100 proteins for the IFN-β recognition and modulation of its activity are probably related to the progression of various cancers and other diseases, as indicated by bioinformatics analysis.

## 2. Results and Discussion

### 2.1. Conformation-Dependent Interactions between IFN-β and Specific S100 Proteins

IFN-β (recombinant human interferon beta-1a) was immobilized on surface of SPR sensor chip by amine coupling and injections of recombinant human S100A1/A4/A6/A7/A8/A9/A10/A11/A12/A13/A14/A15/B solutions were carried out at 25 °C, pH 7.4 running buffer with 1 mM CaCl_2_/EDTA. While no effects were observed for the Ca^2+^-depleted S100 proteins at protein concentrations of 1 µM, and 5 μM solutions of Ca^2+^-loaded S100A7/A8/A9/A10/A11/A12/A13/A14/A15/B proteins, the SPR sensograms for Ca^2+^-bound S100A1/A4/A6 proteins (0.25/0.5–5 µM) exhibited an S100 concentration-dependent pattern (Figure 1). The resulting kinetic SPR data are well approximated by the heterogeneous ligand model [1] (Figure 1) with the lowest equilibrium dissociation constants, *K_d_*, of 41 nM to 227 nM (Table 1). The analogous SPR experiments for IFN-α (recombinant human interferon alfa-2b; sequence identity with IFN-β of 31%) used as a ligand did not reveal any S100-induced effects (see Figure 1 for S100A1/A4/A6), thereby confirming selective nature of the IFN-β recognition by S100A1/A4/A6 proteins.

Considering that affinity of monomeric S100P towards IFN-β and IL-11 differs from that of the S100P dimer [30,41], we performed the analogous SPR experiment for S100A1/A4/A6 proteins, but under their immobilization on the surface of the chip, ensuring S100 representation as a monomer. The SPR sensograms for 10/40/80—40/160 nM IFN-β used as an analyte exhibited a concentration-dependent association-dissociation pattern (Figure 2). The resulting data are adequately approximated by the heterogeneous ligand model [1] with the lowest *K_d_* values of 0.11–1.0 nM (Table 1). Hence, the conversion of Ca^2+^-loaded S100A1/A4/A6 into the monomeric forms increases their affinity to IFN-β by at least two orders of magnitude, as previously observed for S100P [30]. Notably, to the best of our knowledge, these are the strongest interactions reported for S100A1/A4/A6 proteins to date. Similar affinities have been reported for interaction of Ca^2+^-bound S100A6 with sRAGE or is domains (*K_d_* of 0.5–0.6 nM [42]), and for binding of Ca^2+^-loaded S100A4 to Tag7 (apparent *K_d_* of 1 nM [43]).

The fact that the SPR data for 0.5 µM solutions of S100A1/A4/A6 proteins used as an analyte (Figure 1) are well-described within the heterogeneous ligand model [1] with the *K_d_* values drastically different from those estimated using IFN-β as an analyte (Table 1) indicates that these data correspond to multimeric forms of S100A1/A4/A6. Hence, dimer dissociation constants for Ca^2+^-loaded states of S100A1/A4/A6 proteins do not exceed 0.5 µM. This estimate for S100A4 protein is somewhat lower than the previously reported S100A4 dimer dissociation constant of 0.7–1.0 µM [44].

We verified the S100A1/A4/A6 binding to IFN-β by chemical crosslinking. To this end, mixtures of IFN-β (0–5 μM) with S100A1/A4/A6 (S100 to IFN-β molar ratios of 2:0, 0:1, 0.5:1, 1:1, 2:1) were treated with EDAC/sulfo-NHS at 25 °C for 1.5 h, and the resulting cross-linked protein species were analyzed by SDS-PAGE (Figure 3). The cross-linked 5 μM IFN-β sample is mostly monomeric, but contains some oligomeric forms. The cross-linked 10 μM S100A1 is also mainly monomeric, likely due to its poor susceptibility to the crosslinking (Figure 3A). The mixture of IFN-β and Ca^2+^-free S100A1 reveals lowered content of monomeric IFN-β and S100A1, and appearance of the bands above 30 kDa, corresponding to the S100A1—IFN-β complex. Meanwhile, the bands of 10 μM Ca^2+^-bound S100A1 disappear upon addition of 5 μM IFN-β, along with accumulation of the high-molecular-weight bands. The decline in the S100A1 level causes gradual recovery of the bands corresponding to IFN-β. These effects evidence formation of the S100A1—IFN-β complex, with some preference to Ca^2+^-loaded S100A1. Similar effects are observed for S100A4 protein (Figure 3B). The cross-linked 10 μM S100A4 contains trimer and higher order multimers. Addition of 5 μM IFN-β induces disappearance of trimeric Ca^2+^-free S100A4 and accumulation of its monomeric and tetrameric forms. The trimer of 10 μM Ca^2+^-S100A4 disappears upon addition of 5 μM IFN-β. The decrease in S100A4 concentration restores the band of monomeric IFN-β. Thus, the data indicate Ca^2+^-dependent S100A4—IFN-β interaction. Similarly to S100A4, the cross-linked 10 μM S100A6 contains dimer and higher order multimers (Figure 3C). While addition of 5 μM IFN-β to Ca^2+^-free S100A6 is not accompanied by major effects, Ca^2+^-loaded S100A6 loses its dimeric form upon addition of IFN-β. The lowering of S100A6 concentration recovers the bands of IFN-β. Overall, the crosslinking experiments evidence Ca^2+^-dependent IFN-β—S100A1/A4/A6 interactions with equilibrium dissociation constants reaching a micromolar level.

Summing up, among the fourteen S100 proteins studied (including S100P [30]) IFN-β binds S100A1/A4/A6/P proteins, depending on their conformation, with a preference for the monomeric Ca^2+^-loaded state. The strong calcium dependence of these interactions is likely due to the Ca^2+^-induced exposure of hydrophobic surfaces of S100 proteins, favoring target binding [45], which is typical for calcium sensor proteins [46]. Considering that serum S100A1/A4/A6/P levels under pathological conditions reach 3–11 nM (up to 95 nM for S100A4) [47,48,49,50], which exceeds the lowest *K_d_* values for their monomeric forms (0.11–1.0 nM—see Table 1 and ref. [41]), S100A1/A4/A6/P interactions with IFN-β could be of physiological significance. Since therapeutic use of IFN-β-1a raises its serum concentration up to 20 pM (100 IU/mL [51]), which is orders of magnitude less than serum S100A1/A4/A6/P concentrations (above 0.3–1.8 nM [47,48,49,50]), the IFN-β binding cannot affect their functioning. Meanwhile, the S100A1/A4/A6/P binding could modify IFN-β activities.

### 2.2. Modulation of IFN-β Cytotoxicity towards MCF-7 Cells by S100A1/A4/A6

To probe the ability of exogenous S100A1/A4/A6 to affect the IFN-β-induced cellular processes, we studied the influence of IFN-β, various S100 at different concentrations and IFN-β—S100 mixtures on viability of MCF-7 cells by crystal violet assay, as previously described for S100P [30] (Figure 4, Figure 5 and Figure 6). In line with the previous reports [52,53,54], 200 pM IFN-β decreases viability of the cells by 17–32%. While 25–200 nM of S100A1 alone (close to the *K_d1_* value shown in Table 1) does not affect the viability of MCF-7 cells (Figure 4A), the combined application of S100A1 and 200 pM IFN-β exhibits gradual inhibition of the IFN-β-induced effect upon increases in the S100A1 concentration (Figure 4B). Hence, S100A1-bound IFN-β has partially suppressed cytotoxicity against the MCF-7 cells. Similarly, the increase in S100A4 concentration from 5 nM to 200 nM is accompanied by gradual inhibition of the IFN-β-induced suppression of viability of MCF-7 cells (Figure 5B), whereas S100A4 alone is inactive against these cells (Figure 5A). Furthermore, in these cases, we see a correlation between our estimates of IFN-β affinities to the S100 proteins and the results of the cellular assays. In fact, Figure 4 and Figure 5 show that the half-transitions for the MCF-7 viability are observed at S100A1/A4 concentrations of 50 nM, which corresponds to the S100A4-IFN-β equilibrium dissociation constants (*K_d_*) of 41/47 nM, and in-between the estimates of *K_d_* values for IFN-β interactions with monomeric S100A4 (1/105 nM). Meanwhile, exogenous S100A6 does not induce well-defined changes in viability of MCF-7 cells (Figure 6), which indicates that structural peculiarities of the IFN-β—S100A6 complex differ from those for IFN-β—S100A1/A4 complexes.

Examination of the type I interferon ternary complexes with IFNAR1/IFNAR2 [55] shows efficient shielding of the interferon molecules from the solvent. Therefore, S100A1/A4 binding to IFN-β likely sterically hinders the assembly of the ternary complex. The analogous inhibitory effect was previously reported for S100P [30] and for S100B with regard to the FGF2-induced increase in proliferation of MCF-7/MDA-MB468 cells [56].

The half-transition S100A1/A4 concentration in the MCF-7 viability assays (Figure 4 and Figure 5) of 50 nM exceeds the serum S100A1/A4 levels under pathological conditions of 5–11 nM (may reach 95 nM for S100A4) [47,48]. Meanwhile, local S100A1/A4 concentrations within sites of the pathological processes are expected to be substantially higher, thereby favoring development of the cellular effects observed under the in vitro conditions.

### 2.3. Human Diseases Associated to Dysregulation of IFN-β and S100A1/A4/A6

Since S100A1/A4/P proteins clearly affect IFN-β signaling (Figure 4 and Figure 5 and ref. [30]), the S100—IFN-β interactions could be physiologically relevant. IFN-β is secreted in response to various pathological conditions [3], which could be accompanied by the elevation of the levels of S100 proteins, promoting their interaction with IFN-β. In search of the diseases associated with IFN-β and S100A1, S100A4 or S100A6 proteins, we have analyzed DisGeNET and Open Targets Platform (“OTP”) databases of human diseases (see Table 2).

DisGeNET and OTP contain 28 and 94 diseases related simultaneously to IFN-β and S100A1 protein (Appendix A, respectively). Consideration of the OTP entries with the association scores for the both proteins above 0.1 reveals 12 diseases, which cover a wide spectrum of maladies (Appendix A). Similarly, DisGeNET and OTP reveal 104 and 202 diseases associated with the IFN-β and S100A4 protein (Appendix A, respectively). The association scores exceed 0.1 for the 27 diseases (Appendix A), which correspond to various pathologies, including numerous neoplasms and cancers. Finally, IFN-β and S100A6 protein are related to 58 and 150 diseases, according to DisGeNET and OTP databases (see Appendix A), respectively. The association scores exceed 0.1 for the 8 diseases (Appendix A), classified as neoplasms, lung, and nervous system diseases. IFN-β, S100A1, S100A4, and S100A6 proteins share 18 and 53 associated diseases, according to DisGeNET and OTP (Appendix A), respectively. The association scores exceed 0.1 for neoplasm, cancer, carcinoma, lung carcinoma and nervous system disease (Appendix A). Inclusion of S100P into this analysis shows 18 diseases associated with deregulated IFN-β, S100A1, S100A4, S100A6 and S100P proteins, according to the OTP database (Appendix A); the association scores exceed 0.1 for neoplasm, cancer and carcinoma (Appendix A). S100A1/A4/A6/P expression in cancer exhibits bidirectional changes, depending on the protein and cancer type [3,38].

The elevated serum levels of the S100 proteins are expected to favor their interaction with IFN-β, thereby affecting antitumor activity of the latter, considered as a promising agent for cancer treatment. The example of MCF-7 cells points out that antitumor activity of IFN-β is likely to be suppressed by the S100A1/A4/P proteins (Figure 4 and Figure 5 and ref. [30]).

### 2.4. Intrinsic Disorder Predisposition of Human S100 Proteins

To see whether differences between the capabilities of S100 proteins to interact with IFN-β are somehow related to their intrinsic disorder predispositions, we generated intrinsic disorder profiles for S100A1/A4/A6/P that are capable of IFN-β binding and compared them with the disorder profiles of S100B/A7/A8/A9/A10/A11/A12/A13/A14/A15 which do not bind IFN-β (Figure 7).

This comparison revealed a noticeable difference between these two groups of S100 proteins, where the IFN-β binders were characterized by a remarkable similarity of their per-residue disorder predispositions (Figure 7A), whereas the disorder profiles of non-binders were highly diversified within a group (Figure 7B) and were noticeably different from the profiles of binders. The only exception from this regularity is S100B, whose disorder profile was rather similar to those of IFN-β binders. Likely, this can be explained considering evolutionary relations between the proteins, where S100B showed high sequence similarity to S100A1 and S100P (see Appendix A). On the other hand, although according to the results of the multiple sequence alignment (see Appendix A) S100A4 and S100A6 are evolutionary close to S100A13 and S100A14, the disorder profiles of S100A4 and S100A6 are quite different from those of S100A13 and S100A14 (cf. Figure 7A,B). Curiously, although S100B cannot bind IFN-β, it still belongs to the group of “promiscuous” S100 proteins capable of specific interaction with multiple partners [57]. These observations suggest that the peculiarities of the distribution of intrinsic disorder predisposition within the amino acid sequences of S100 proteins might be related to their binding promiscuity and may play a role in interaction with IFN-β.

### 2.5. Modeling of S100—IFN-β Complexes

Examination of tertiary structures of S100A1/A4/A6/P complexes with their targets (Figure 8) reveals that amino acids of the “hinge” region between helices II and III and helix IV of these S100 proteins are involved in target recognition (highlighted in bold). These regions were used for modelling of S100A1/A4/A6/P—IFN-β complexes. Considering that the available tertiary structures of S100A1 and S100A6 proteins complexed with their targets (PDB entries 2K2F, 2KBM, 4P2Y and 2JTT) contain non-human S100 proteins, tertiary structures of human S100A1 and S100A6 proteins (2LP3 and 1K9K, respectively) were used for further modelling. Since S100A1/A4/P proteins are able to prevent IFN-β signaling (Figure 4 and Figure 5 and ref. [30]), indicating that they compete with IFNAR1/IFNAR2 for the same IFN-β site(s), the probable interacting surface of IFN-β was taken from the tertiary structure of its complex with IFNAR1 (PDB entry 3WCY): helices B (Y60, Q64 and A68), C (A89, Y92 and N96) and D (S119, K123 and G127) of mouse IFN-β. The contact amino acids were mapped onto the structure of human IFN-β (PDB entry 1AU1). Modeling of tertiary structures of complexes between human S100A1/A4/A6/P dimer and human IFN-β using GRAMM-X protein-protein docking software v.1.2.0 [58] predicts (Figure 8B) that IFN-β helices B, C and D bind to the “cleft” formed by the “hinge”, helix III and the C-terminal half of helix IV of the primary interacting S100 monomer (shown in green in Figure 9).

However, while the second molecule within the S100P dimer lacks interactions with IFN-β, the second monomer in S100A1/A4/A6 is predicted to bind IFN-β in a S100-dependent manner (Figure 8B): S100A4 binds IFN-β by the N-terminal half of helix I, S100A1 contacts IFN-β molecule via the loop region between helices III and IV, whereas S100A6 binds IFN-β by the same loop region and the N-terminal half of helix IV. The modeling demonstrates the possibility that S100 dimer binds IFN-β molecule via the surface, involved into interactions of the latter with IFNAR1. Notably, Ca^2+^-binding loops of the S100 monomers (orange-colored in Figure 9), except for S100A4 protein, lie close to the interacting interface, which could confer calcium sensitivity to the S100—IFN-β interaction, in accord with experimental observations (Figure 1). Furthermore, the differences in relative orientations of S100A1/A4/A6/P molecules within the models of complexes with IFN-β could explain their different abilities to inhibit the IFN-β-induced suppression of MCF-7 cell viability (Figure 4, Figure 5 and Figure 6 and ref. [30]).

## 3. Materials and Methods

### 3.1. Materials

Human interferon beta-1a produced in CHO-K1 cells (IFN-β) was purchased from Merck (Rebif^®^). Human interferon alfa-2b produced in *E. coli* cells (IFN-α) was from European Directorate for the Quality of Medicines (CRS I0320301). Human S100B was expressed in *E. coli* and purified as described in ref. [61]. Mouse ubiquitin specific peptidase 2 (Usp2) was prepared as described in ref. [62]. Restriction enzymes were from Thermo Scientific^TM^ (Waltham, MA, USA). Hen egg white lysozyme, DNase I were from Sigma-Aldrich Co. (St. Louis, MO, USA).

Human S100A1, S100A4, S100A6, S100A7, S100A8, S100A9, S100A10, S100A11, S100A12, S100A13, S100A14, S100A15 (S100A7A) genes (Swiss-Prot entries P23297, P26447, P06703, P31151, P05109, P06702, P60903, P31949, P80511, Q99584, Q9HCY8, Q86SG5, respectively) were obtained from DNASU Plasmid Repository (https://dnasu.org/DNASU/Home.do): clones HsCD00286778, HsCD00343150, HsCD00343132, HsCD00664846, HsCD00004369, HsCD00516086, HsCD00021287, HsCD00000028, HsCD00403982, HsCD00674114, HsCD00334043, HsCD00302486, respectively.

Tricine and sodium acetate were purchased from Amresco. Phosphate, Tris, glycine, sodium acetate, imidazole, HEPES, ammonium sulfate, DTT, EGTA, NaOH, SDS and glycerol were from PanReac AppliChem. IPTG, PMSF and sodium chloride were from Helicon (Moscow, Russia). CaCl_2_, EDTA and TWEEN 20 were bought from Sigma-Aldrich Co.

TOYOPEARL^®^ SuperQ-650M and Phenyl-650M resins were purchased from Tosoh Bioscience. HiPrep™ 26/60 Sephacryl^®^ S-100 HR, Superdex 75 10/300 GL columns, DEAE Sephacel^TM^ and SP Sepharose^®^ Fast Flow media were from GE Healthcare. Bio-Scale™ Mini Profinity™ IMAC cartridges, Profinity™ IMAC resin, ProteOn™ GLH sensor chip, amine coupling kit, EDAC and sulfo-NHS were from Bio-Rad Laboratories, Inc. (Hercules, CA, USA).

MCF-7 cell line was from European Collection of Authenticated Cell Cultures. PBS, DMEM/F12 medium, FBS, GlutaMAX™, penicillin, streptomycin and trypsin-EDTA were from Gibco. Crystal violet and paraformaldehyde were from Sigma-Aldrich Co.

Protein concentrations were measured spectrophotometrically using molar extinction coefficients at 280 nm calculated according to ref. [63] (refer to Appendix A).

### 3.2. Construction of Plasmids

Human S100A1, S100A4, S100A6, S100A7, S100A8, S100A11, S100A12, S100A14 and S100A15 genes were cloned into pHUE (Histidine-tagged Ubiquitin Expression) vector [64] between *Sac*II and *Hin*dIII restriction sites. Human S100A9, S100A10 and S100A13 genes were cloned into pET-11a vector (Novagen^®^, Merck) between *Nde*I and *Bam*HI restriction sites.

### 3.3. Expression and Purification of S100 Proteins

Samples of recombinant human S100A1, S100A4, S100A6, S100A11 and S100A12 proteins were prepared as follows. *E. coli* BL21 (DE3) cells containing pLacIRARE plasmid were transformed with the pHUE-S100 plasmid. The cells were grown at 37 °C in 1 L of 2YT medium with 100 µg/mL ampicillin, shaking at 200 rpm, until optical density at 600 nm reached 1 AU. Expression of the ubiquitin-S100 chimera was induced by 0.5 mM IPTG. The cells were grown at 25 °C for 4 h, harvested by centrifugation at 5000× *g* for 15 min at 4 °C, resuspended in 30 mL of lysis buffer (50 mM phosphate, 5 mM imidazole, 1 mM PMSF, 1 mM 2-ME, 1 M NaCl, 0.1% TWEEN 20, 0.1 mg/mL hen egg white lysozyme, pH 8.0) and disintegrated using a French press. The lysate was centrifuged at 25,000× *g* for 40 min at 4 °C. The supernatant was loaded onto 5 mL Bio-Scale™ Mini Profinity™ IMAC Ni-charged column. The resin was washed with 50 mL of the lysis buffer. The protein was eluted with 10 mM phosphate, 300 mM imidazole, 1 mM 2-ME, 150 mM NaCl, pH 7.5 buffer. The fractions containing ubiquitin-S100 chimera were joined, dialyzed at 4 °C against 10 mM Tris-HCl, 150 mM NaCl, 1 mM DTT, pH 7.7 buffer and treated with Usp2 peptidase (50–100-fold molar excess of the chimera over the enzyme, 16 h at 37 °C). The protein mixture was dialyzed against 20 mM Tris pH 7.7–8.5 buffer (buffer “A”), loaded onto a 6 mL TOYOPEARL^®^ SuperQ-650M anion exchange column and washed with buffer A. The first peak has been discarded. S100 protein was eluted by a linear gradient of NaCl (0–1.5 M) in buffer A (50 mL; flow rate of 1 mL/min). The collected S100 protein was further purified using a HiPrep™ 26/60 Sephacryl^®^ S-100 HR gel filtration column equilibrated with PBS (flow rate of 1 mL/min). The purified protein was dialyzed at 4 °C against 1:1 (*v/v*) PBS-glycerol mixture with 1 mM DTT and stored at −20 °C.

Samples of human S100A7, S100A8 and S100A15 proteins were similarly prepared. BL21(DE3)-RIL cells were transformed with pHUE-S100 plasmid and grown at 37 °C in 1 L of 2YT medium with 100 µg/mL ampicillin, shaking at 200 rpm, until optical density at 600 nm reached 0.7 AU. Expression of the ubiquitin-S100 chimera was induced by 1 mM IPTG. The cells were grown at 37 °C for 4 h, harvested by centrifugation at 5000× *g* for 15 min at 4 °C, resuspended in 30 mL of lysis buffer (20 mM Tris-HCl, 0.5 mM imidazole, 1 mM PMSF, 1 mM 2-ME, 0.5 M NaCl, pH 8.0) and disintegrated using a French press. The lysate was centrifuged at 28,000× *g* for 1 h at 4 °C. The supernatant was loaded onto 5 mL Bio-Scale™ Mini Profinity™ IMAC column. The resin was washed with the lysis buffer without PMSF. The protein was eluted with 20 mM Tris-HCl, 500 mM imidazole, 1 mM 2-ME, 150 mM NaCl, pH 8.0 buffer. The fractions containing ubiquitin-S100 chimera were treated with Usp2 peptidase (50–100-fold molar excess of the chimera over the enzyme, 16 h at 37 °C) in the presence of 15 mM 2-ME. The protein mixture was dialyzed against buffer “A” (20 mM Tris-HCl, 1 mM EDTA, 1 mM 2-ME, pH 8.0) or buffer “B” (10 mM glycine-NaOH, 1 mM 2-ME, pH 9.0), for S100A7/S100A8 and S100A15 proteins, respectively. The dialyzed protein sample was loaded onto a TOYOPEARL^®^ SuperQ-650M column (0.9 × 10 cm). S100 protein was eluted by a linear gradient of NaCl (0–0.6 M) in the respective buffer (A or B; flow rate of 2 mL/min). The collected S100 protein was further purified using a HiPrep™ 26/60 Sephacryl^®^ S-100 HR column equilibrated with 20 mM Tris-HCl, 1 mM 2-ME, 150 mM NaCl, pH 8.0 buffer (flow rate of 1 mL/min). The purified protein was concentrated, dialyzed at 4 °C against 20 mM Tris-HCl, 1 mM DTT, 150 mM NaCl, pH 8.0 buffer, and stored with 10% glycerol at −70 °C.

BL21(DE3)-RIL cells were transformed with pHUE-S100A14 plasmid and grown at 37 °C in 1 L of 2YT medium with 100 µg/mL ampicillin, shaking at 170 rpm, until optical density at 600 nm reached 1.5 AU. Expression of the ubiquitin-S100A14 chimera was induced by 1 mM IPTG. The cells were grown at 37 °C for 3 h, harvested by centrifugation at 5000× *g* for 15 min at 4 °C, resuspended in 30 mL of lysis buffer (50 mM Tris-HCl, 1 mM PMSF, 1 mM EDTA, 1 mM 2-ME, 1 M NaCl, 0.05% TWEEN 20, pH 8.0) and disintegrated using a French press. The lysate was centrifuged at 20,000× *g* for 1 h at 4 °C. The supernatant was loaded onto 5 mL Bio-Scale™ Mini Profinity™ IMAC column. The resin was washed with 20 mM Tris-HCl, 5 mM imidazole, 1 mM 2-ME, 1 M NaCl, pH 8.1 buffer. The protein was eluted with 20 mM Tris-HCl, 300 mM imidazole, 1 mM 2-ME, 150 mM NaCl, pH 8.1 buffer. The fractions containing ubiquitin-S100A14 chimera were collected, dialyzed at 4 °C against 50 mM Tris-HCl, 5 mM 2-ME, 300 mM NaCl, pH 8.2 buffer and incubated with Usp2 peptidase overnight at 37 °C. The protein solution was cleared by centrifugation at 20,000× *g* for 30 min at 4 °C and incubated with 5 mL of Profinity™ IMAC resin for 1 h at room temperature under gentle stirring. The unbound S100A14 protein was collected, dialyzed at 4 °C against 50 mM Tricine-NaOH, 5 mM DTT, 300 mM NaCl, 5% glycerol, pH 7.5 (buffer A) and further purified using Superdex 75 10/300 GL gel filtration column equilibrated with buffer A (flow rate of 0.65 mL/min). The purified protein was concentrated and stored with 10% glycerol at −70 °C.

BL21(DE3)pLysS cells were transformed with pET-11a-S100A9 plasmid and grown at 37 °C in 1 L of 2YT medium with 100 µg/mL ampicillin, shaking at 200 rpm, until optical density at 600 nm reached 0.6 AU. S100A9 expression was induced by 1 mM IPTG. The cells were grown at 37 °C for 4 h, harvested by centrifugation at 5000× *g* for 15 min at 4 °C, resuspended in 30 mL of lysis buffer (50 mM Tris-HCl, 1 mM PMSF, 2 mM EDTA, 1 mM DTT, pH 7.5) and disintegrated using a French press. The lysate was centrifuged at 28,000× *g* for 1 h at 4 °C, followed by incubation of the supernatant with 50% ammonium sulfate on ice under stirring for 30 min and centrifugation at 20,000× *g* for 30 min at 4 °C. The supernatant was dialyzed at 4 °C against distilled water with 1 mM 2-ME, followed by dialysis against 20 mM Tris-HCl, 1 mM EDTA, 1 mM 2-ME, pH 8.0 (buffer A). The dialyzed protein was loaded onto a TOYOPEARL^®^ SuperQ-650M column (3 × 6 cm) equilibrated with buffer A. S100A9 protein was eluted by a linear gradient of NaCl (0–0.6 M) in buffer A (flow rate of 2 mL/min). The fractions containing S100A9 were collected, dialyzed at 4 °C against distilled water with 1 mM 2-ME, followed by dialysis against 20 mM sodium acetate, 1 mM 2-ME, pH 5.0 (buffer B). The protein solution was loaded onto a SP Sepharose^®^ Fast Flow column (0.9 × 7.5 cm) equilibrated with buffer B. S100A9 was eluted by a linear gradient of NaCl (0–0.7 M) in buffer B (flow rate of 1 mL/min) and further purified using a HiPrep™ 26/60 Sephacryl^®^ S-100 HR column equilibrated with 20 mM Tricine-NaOH, 1 mM 2-ME, 150 mM NaCl, pH 7.5 buffer (flow rate of 1 mL/min). The purified protein was concentrated, dialyzed at 4 °C against 20 mM Tricine-NaOH, 1 mM DTT, 150 mM NaCl, pH 7.5 buffer and stored with 10% glycerol at −70 °C.

BL21(DE3)pLysS cells were transformed with the pET-11a-S100A10 plasmid and grown at 37 °C in 1 L of 2YT medium with 100 µg/mL ampicillin, shaking at 200 rpm, until optical density at 600 nm reached 0.6 AU. S100A10 expression was induced by 1 mM IPTG. The cells were grown at 37 °C for 4 h, harvested by centrifugation at 5000× *g* for 15 min at 4 °C, resuspended in 30 mL of lysis buffer (50 mM Tris-HCl, 1 mM PMSF, 2 mM EDTA, 1 mM 2-ME, pH 7.5) and disintegrated using a French press. The lysate was centrifuged at 20,000× *g* for 30 min at 4 °C, followed by loading of the supernatant onto a 30 mL DEAE Sephacel^TM^ weak anion exchanger column equilibrated with 50 mM Tris-HCl, 10 mM 2-ME, 20 mM NaCl, pH 7.5 (buffer A). S100A10 protein was eluted by buffer A (flow rate of 2 mL/min), dialyzed at 4 °C against distilled water with 10 mM 2-ME, followed by dialysis against 50 mM sodium acetate, 10 mM 2-ME, 0.5 mM EDTA, 0.5 mM EGTA, 20 mM NaCl, pH 5.6 (buffer B). The protein solution was loaded onto SP Sepharose^®^ Fast Flow column (0.9 × 7.5 cm) equilibrated with buffer B. S100A10 was eluted by a linear gradient of NaCl (0.1–1 M) in buffer B (flow rate of 3 mL/min), and further purified using HiPrep™ 26/60 Sephacryl^®^ S-100 HR column equilibrated with 20 mM Tricine-NaOH, 1 mM 2-ME, 150 mM NaCl, pH 7.5 buffer (flow rate of 1 mL/min). The purified protein was concentrated and stored in 20 mM Tricine-NaOH, 1 mM 2-ME, 150 mM NaCl, pH 7.5 buffer with 10% glycerol at −70 °C.

BL21(DE3) cells containing pLacIRARE plasmid were transformed with pET-11a-S100A13 plasmid and grown at 37 °C in 1 L of 2YT medium with 100 µg/mL ampicillin, shaking at 170 rpm, until optical density at 600 nm reached 0.9 AU. S100A13 expression was induced by 0.5 mM IPTG. The cells were grown at 37⁰ C for 4 h, harvested by centrifugation at 5000× *g* for 15 min at 4 °C, resuspended in 30 mL of lysis buffer (20 mM Tris-HCl, 1 mM PMSF, 1 mM EDTA, 2 mM 2-ME, 10 mM MgCl_2_, 300 mM KCl, pH 7.4 with 1 µg/mL DNase I) and disintegrated using a French press. The lysate was centrifuged at 20,000× *g* for 30 min at 4 °C, followed by incubation of the supernatant with 30% ammonium sulfate on ice under stirring for 20 min, and centrifugation at 15,000× *g* for 15 min at 1 °C. 2 mM CaCl_2_ was added to the supernatant, and the mixture was loaded onto a TOYOPEARL^®^ Phenyl-650M hydrophobic interaction chromatography column (2.5 × 4.8 cm) equilibrated with 20 mM Tris-HCl, 2 mM 2-ME, 300 mM KCl, pH 7.4 (buffer A) with 2 mM CaCl_2_ and 30% ammonium sulfate. S100A13 was eluted by buffer A with 1 mM EDTA (flow rate of 3 mL/min). The fractions containing S100A13 were collected, dialyzed at 4 °C against 20 mM Tris-HCl, 10 µM CaCl_2_, 2 mM 2-ME, pH 8.0 (buffer B) and loaded onto a TOYOPEARL^®^ SuperQ-650M column (0.9 × 7.5 cm) equilibrated with buffer B. S100A13 was eluted by a linear gradient of KCl (0–300 mM) in buffer B (flow rate of 2 mL/min). The fractions containing S100A13 were collected, followed by addition of 100 mM sodium acetate and pH adjustment to 4.5. The protein sample was dialyzed at 4 °C against 20 mM sodium acetate, 2 mM 2-ME, pH 4.5 (buffer C) and loaded onto a SP Sepharose^®^ Fast Flow column (0.9 × 7.5 cm) equilibrated with buffer C. S100A13 was eluted by a linear gradient of NaCl (0–0.7 M) in buffer C (flow rate of 1 mL/min). The purified protein was dialyzed at 4 °C against 20 mM Tricine-NaOH, 2 mM DTT, pH 7.3 buffer and stored with 10% glycerol at −70 °C.

Homogeneity of S100 protein samples was controlled by SDS-PAGE. Their molecular masses were verified by electrospray ionization mass spectrometry as previously described [65].

### 3.4. Surface Plasmon Resonance Studies

SPR measurements were performed at 25 °C using Bio-Rad ProteOn™ XPR36 instrument generally as described in ref. [30]. Ligand (30–50 μg/mL IFN-β/IFN-α or S100A1/A4/A6) was immobilized on ProteOn™ GLH sensor chip surface (up to 12,000–17,000 resonance units, RUs) by amine coupling. Analyte (0.25–5 μM S100A1, 0.5–5 μM S100A4/A6, 5 μM S100A7/A8/A9/A10/A11/A12/A13/A14/A15/B, or 10–160 nM IFN-β) in a running buffer (10 mM HEPES, 150 mM NaCl, 0.01% TWEEN 20, pH 7.4 buffer with 1 mM CaCl_2_/EDTA) was passed over the chip, followed by flushing the chip with the running buffer. The sensor chip surface was regenerated by 20 mM EDTA pH 8.0 solution. The data were globally fitted according to a heterogeneous ligand model, which assumes the existence of two populations of the ligand (L_1_ and L_2_) that bind single analyte molecule (A):*K_d1_*       *K_d2_*L_1_ + A ↔ L_1_A;  L_2_ + A ↔ L_2_A  *k_d1_*       *k_d2_*(1)
where *K_d_* and *k_d_* refer to equilibrium and kinetic dissociation constants, respectively.

One might ask a question on the nature of the L_1_ and L_2_ ligands in the case of the S100-IFN-β interaction. Since the ligand is chemically linked to the surface of the SPR chip via amino groups, it is possible that the ligand can be exposed to the analyte in several preferential conformations, differing in their ability to recognize the analyte. Some of the ligand conformations may preclude the interaction with the analyte due to sterical hindrance, while others are characterized by an “open” conformation. It is likely that L_1_ and L_2_ correspond to “open” conformations of the ligand with a different affinity to the analyte. They could correspond to different analyte-binding sites or different conformations of the same site. Although this is very typical for SPR experiments, it is rarely possible to distinguish between these scenarios.

### 3.5. Chemical Crosslinking

S100 stock solutions were incubated with 20 mM DTT and 20 mM EDTA for 1 h, followed by 10-fold concentration in the buffer (see below) using centrifugal concentrators with a molecular weight cutoff of 3.5 kDa. The concentrated samples were diluted with the same buffer to the parent volume and reconcentrated (6 times in total). Crosslinking of protein samples (5 μM IFN-β with S100A1/A4/A6 added up to S100 to IFN-β molar ratio of 2:0, 0:1, 0.5:1, 1:1, 2:1) with 40 mM EDAC and 10 mM sulfo-NHS was performed at 25 °C for 1.5 h (10 mM HEPES, 150 mM NaCl, pH 7.4, either with 1 mM CaCl_2_ for Ca^2+^-bound S100 or without addition of CaCl_2_ for Ca^2+^-depleted S100). The reaction was quenched by addition of SDS-PAGE sample loading buffer. The samples (20 μL) were analyzed by SDS-PAGE (15%) using silver staining.

### 3.6. Cell Viability Studies

MCF-7 cells were cultured in DMEM/F12 medium supplemented with 10% FBS, 2 mM GlutaMAX™, 1 mM sodium pyruvate, 100 IU/mL penicillin and 100 μg/mL streptomycin at 37 °C in humidified atmosphere with 5% CO_2_. The cells were harvested by trypsinization and seeded in 96-well microplates at concentration of 200 cells per well, 100 µL. On the next day, 100 µL of preincubated at 37 °C for 1 h solutions of IFN-β (400 pM) or S100A1/A4/A6 (10, 20, 50, 100, 200 or 400 nM) in the growth medium without FBS were added to the wells. Alternatively, 50 µL of the IFN-β (800 pM) and 50 µL of the S100A1/A4/A6 (20, 40, 100, 200, 400 or 800 nM) solutions were added to the wells and 100 µL of the growth medium without FBS were added to the control wells. After four days of cultivation, the cells were fixed with 4% ice-cold formaldehyde solution in PBS at 4 °C for 30 min, washed twice with PBS and stained with 0.5% crystal violet solution in 20% (*v/v*) ethanol for 10 min at room temperature. The plates were flushed with water and drained, followed by addition of 150 µL of 1% SDS to the wells. Absorbance at 596 nm was read by Multiskan^TM^ FC Microplate Photometer (Thermo Scientific^TM^). The data are presented as the means ± standard deviations (*n* = 5–6).

### 3.7. Search of Diseases Associated to IFN-β and S100 Proteins

The data on diseases associated with genes IFNB1 (human IFN-β, UniProt ID P01574) and either S100A1 (human S100A1, UniProt ID P23297), S100A4 (human S100A4, UniProt ID P26447) or S100A6 (human S100A6, UniProt ID P06703) were collected from the human disease databases DisGeNET v7.0 (http://www.disgenet.org) [66] and Open Targets Platform v.20.09 (https://www.opentargets.org) [67] as described in ref. [30]. The records found in DisGeNET were manually curated, and false positive records were removed.

### 3.8. Evaluation of Intrinsic Disorder Predispositions of S100 Proteins

Intrinsic disorder predispositions of S100 proteins analyzed in this study (S100A1/S10A1 (UniProt ID: P23297), S100A4/S10A4 (UniProt ID: P26447), S100A6/S10A6 (UniProt ID: P06703), S100A7/S10A7 (UniProt ID: P31151), S100A8/S10A8 (UniProt ID: P05109), S100A9/S10A9 (UniProt ID: P06702), S100A10/S10AA (UniProt ID: P60903), S100A11/S10AB (UniProt ID: P31949), S100A12/S10AC (UniProt ID: P80511), S100A13/S10AD (UniProt ID: Q99584), S100A14/S10AE (UniProt ID: Q9HCY8), S100A15/S100A7A/A1A7A (UniProt ID: Q86SG5), S100P (UniProt ID: P25815), and S100B (UniProt ID: P04271)) were evaluated by PONDR^®^ VSL2, which is one of the more accurate tools for prediction of intrinsic disorder [68]. The outputs of this tool are represented as real numbers between 1 (ideal prediction of disorder) and 0 (ideal prediction of order). A threshold of ≥0.5 was used to identify disordered residues and regions in query proteins. A protein region was considered flexible if its disorder propensity was in a range from 0.2 to 0.5.

### 3.9. Multiple Sequence Alignments of S100 Proteins

Multiple sequence alignments were conducted for three groups of proteins: (a) S100 proteins capable of IFN-β (S100P, S100A1, S100A4 and S100A6), (b) non-binders (S100B/A7/A8/A9/A10/A11/A12/A13/A14/A15); and (c) all S 100 proteins analyzed in this study were conducted using the Clustal Omega algorithm [69] available at https://www.ebi.ac.uk/Tools/msa/clustalo/. In these analyses, we utilized the default parameters for the alignments and outputs, except to the order in which the sequences appear in the final alignment, where instead of “aligned”, “input” mode was used. Clustal Omega was also used to obtain the Percent Identity Matrices for all the group and to generate evolutionary tree for all human S100 proteins analyzed in this study.

### 3.10. Modeling of S100—IFN-β Complexes

Tertiary structures of proteins were extracted from the Protein Data Bank (PDB) [70]. The models of tertiary structures of the S100—IFN-β complexes were built based on the structures of human IFN-β (chain A of PDB entry 1AU1, X-ray), Ca^2+^-loaded human S100P dimer in complex with V domain of the receptor for advanced glycation end products (RAGE) (chains B and D of PDB entry 2MJW, NMR, conformer 1), Ca^2+^-bound human S100A1 dimer (chains A and B of PDB entry 2LP3, NMR, conformer 1), Ca^2+^-bound human S100A4 dimer complexed with Annexin A2 (chains C and D of PDB entry 5LPU, X-ray) and Ca^2+^-loaded human S100A6 dimer (chains A and B of PDB entry 1K9K, X-ray), using GRAMM-X protein–protein docking software v.1.2.0 [58]. The best docking combination generated by GRAMM-X was chosen considering the sets of interacting atoms and the overall intermolecular orientations found in the aforementioned structures. The interacting surfaces for IFN-β and S100 proteins were taken from the following complexes: murine IFN-β—IFNAR1 (PDB entry 3WCY, X-ray), Ca^2+^-bound human S100P dimer—RAGE (PDB entry 2MJW, NMR), Ca^2+^-bound rat S100A1 dimer complexed with RyRP12 (PDB entry 2K2F, NMR) or TRTK12 (PDB entry 2KBM, NMR), Ca^2+^-loaded human S100A4 dimer in complex with Annexin A2 (PDB entry 5LPU, X-ray) or nonmuscle myosin IIA (PDB entry 2LNK, NMR), Ca^2+^-loaded murine S100A6—human RAGE (PDB entry 4P2Y, X-ray), Ca^2+^-bound rabbit S100A6 dimer—mouse SIP (PDB entry 2JTT, NMR). The contact residues were derived with Ligand-Protein Contacts (LPC) software [60] and mapped onto the structures of the homologous proteins by pairwise structure comparisons, as implemented on DALI server (http://ekhidna2.biocenter.helsinki.fi/dali/ [59]). All additional modeling, superposition and analyses were performed using BIOVIA Discovery Studio (https://www.3dsbiovia.com/). Tertiary structures were drawn with MOLSCRIPT software [71].

## 4. Conclusions

We have found three novel highly specific interactions between members of the two critically important protein families, type I interferons and S100 proteins. The IFN-β—S100A1/A4/A6 interactions are selective, since IFN-β replacement by IFN-α and use of S100A7/A8/A9/A10/A11/A12/A13/A14/A15/B instead of S100A1/A4/A6 did not reveal interactions with the *K_d_* values below 10 µM. Previously, IFN-β recognition was documented also for S100P protein [30]. In all cases, Ca^2+^-loaded monomeric forms of S100 proteins possess maximal affinities to IFN-β, with the *K_d_* values laying in (sub)nanomolar range, which is close to the physiological S100 levels. Unfortunately, tertiary structures of monomeric S100 proteins are unknown due to their very low dimer dissociation constants, which impedes structural modelling of their complexes with IFN-β and limits search of the probable IFN-β-recognizing surfaces conserved within S100A1/A4/A6/P proteins.

It should be noted that the S100 proteins highly specific to IFN-β belong to the group of “promiscuous” S100 proteins, able to bind several ligands with a high affinity, while the S100 proteins exhibiting lack of specificity to IFN-β except for S100B protein are considered as “orphan” binders, which weakly bind to no more than one partner [57]. This fact indicates that the revealed cross-interactions of S100A1/A4/A6/P proteins with IFN-β could be a consequence of the property inherent to the S100 protein family. Furthermore, S100A1/A4/A6 proteins are evolutionarily closely related [72]. In any case, the high-affinity IFN-β interaction with specific S100 proteins is expected to impair its signaling due to sterical prevention of IFNAR1/IFNAR2 binding. Therefore, the inhibitory effect of S100A1/A4/P proteins on IFN-β-induced suppression of cell viability (Figure 4 and Figure 5 and ref. [30]) is likely to be amplified by their concerted action. In this case, directed blocking of these S100 proteins could enhance therapeutic efficiency of IFN-β. This approach is especially relevant to oncological diseases, frequently accompanied by increased S100A1/A4/A6/P expression [3,38].

The S100-cytokine interactions leading to modulation of cytokine activities seem to be a more common phenomenon than previously thought (reviewed in ref. [30]). Considering the vast involvement of cytokines and S100 proteins into the pathogenesis of socially significant diseases, knowledge of details of the novel regulatory function of S100 proteins could serve as a background for the development of innovative therapeutic approaches.

## Figures and Tables

**Figure 1 ijms-21-09473-f001:**
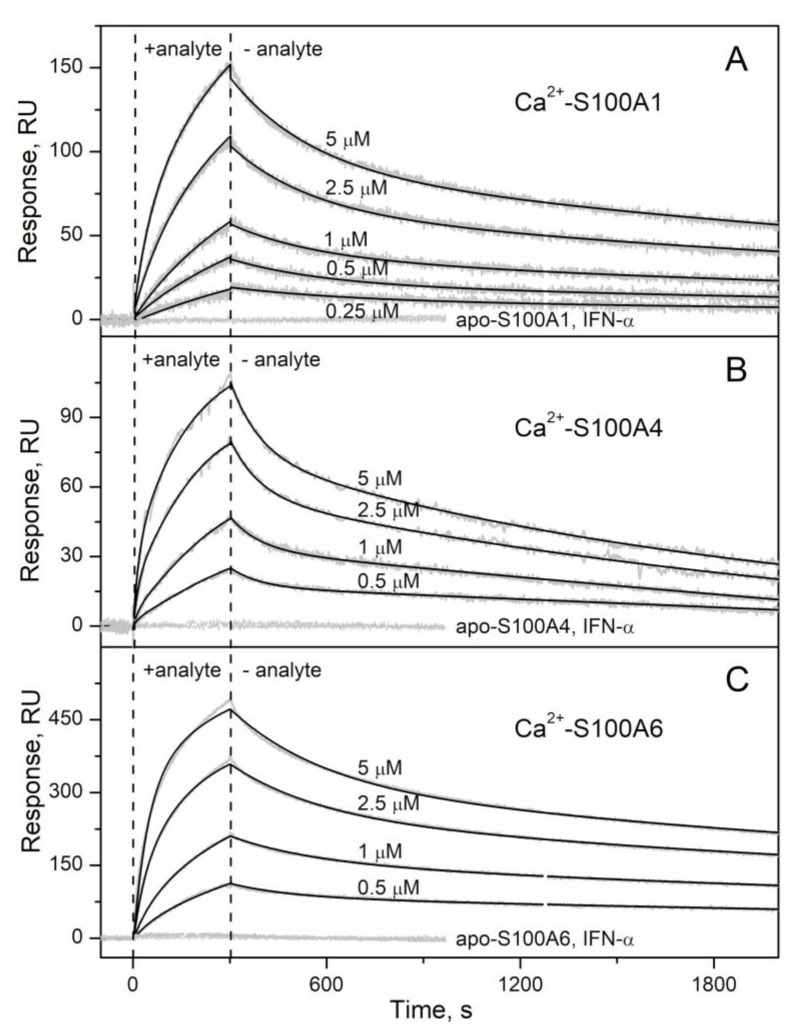
Kinetics of the interaction between IFN-β and Ca^2+^-bound S100 proteins at 25 °C (1 mM CaCl_2_, pH 7.4), monitored by SPR spectroscopy using IFN-β as a ligand and S100A1 (panel **A**), S100A4 (**B**) or S100A6 (**C**) as an analyte (0.25–5 µM). The sensograms for control experiments are also indicated: Ca^2+^-depleted S100 proteins (1 µM; 1 mM EDTA) were used as an analyte; IFN-α served as a ligand. Grey curves are experimental, while black curves are theoretical, calculated according to the heterogeneous ligand model [1] (refer to Table 1 for the fitting parameters).

**Figure 2 ijms-21-09473-f002:**
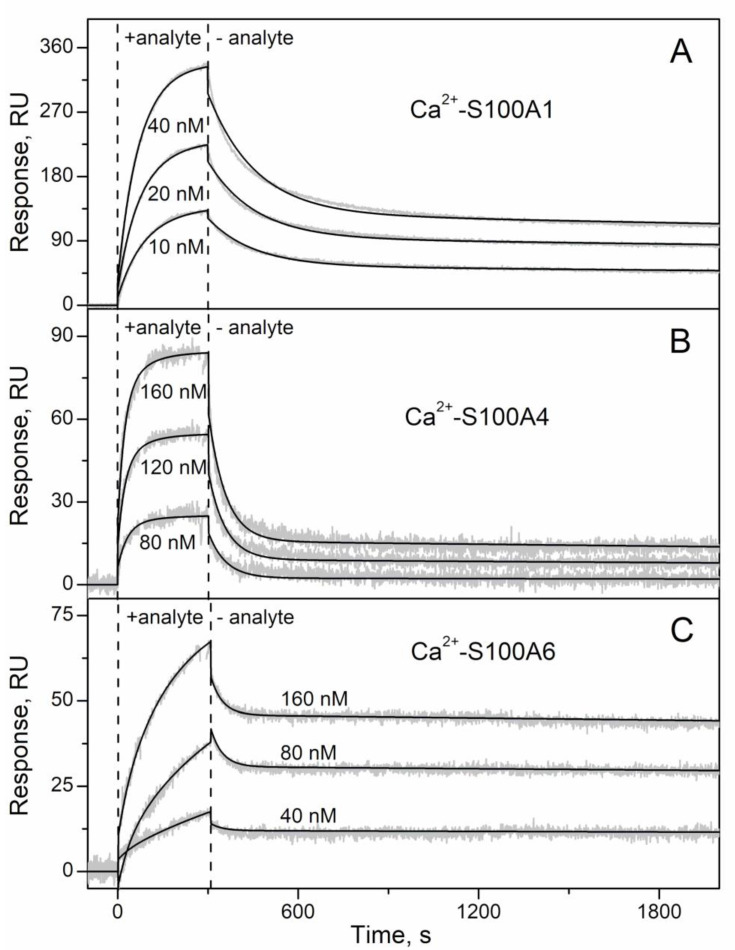
Kinetics of the interaction between IFN-β- and Ca^2+^-bound S100 proteins at 25 °C (1 mM CaCl_2_, pH 7.4), monitored by SPR spectroscopy using IFN-β as a ligand and S100A1 (panel **A**), S100A4 (**B**) or S100A6 (**C**) as an analyte (0.25–5 µM). The sensograms for control experiments are also indicated: Ca^2+^-depleted S100 proteins (1 µM; 1 mM EDTA) were used as an analyte; IFN-α served as a ligand. Grey curves are experimental, while black curves are theoretical, calculated according to the heterogeneous ligand model [1] (refer to Table 1 for the fitting parameters).

**Figure 3 ijms-21-09473-f003:**
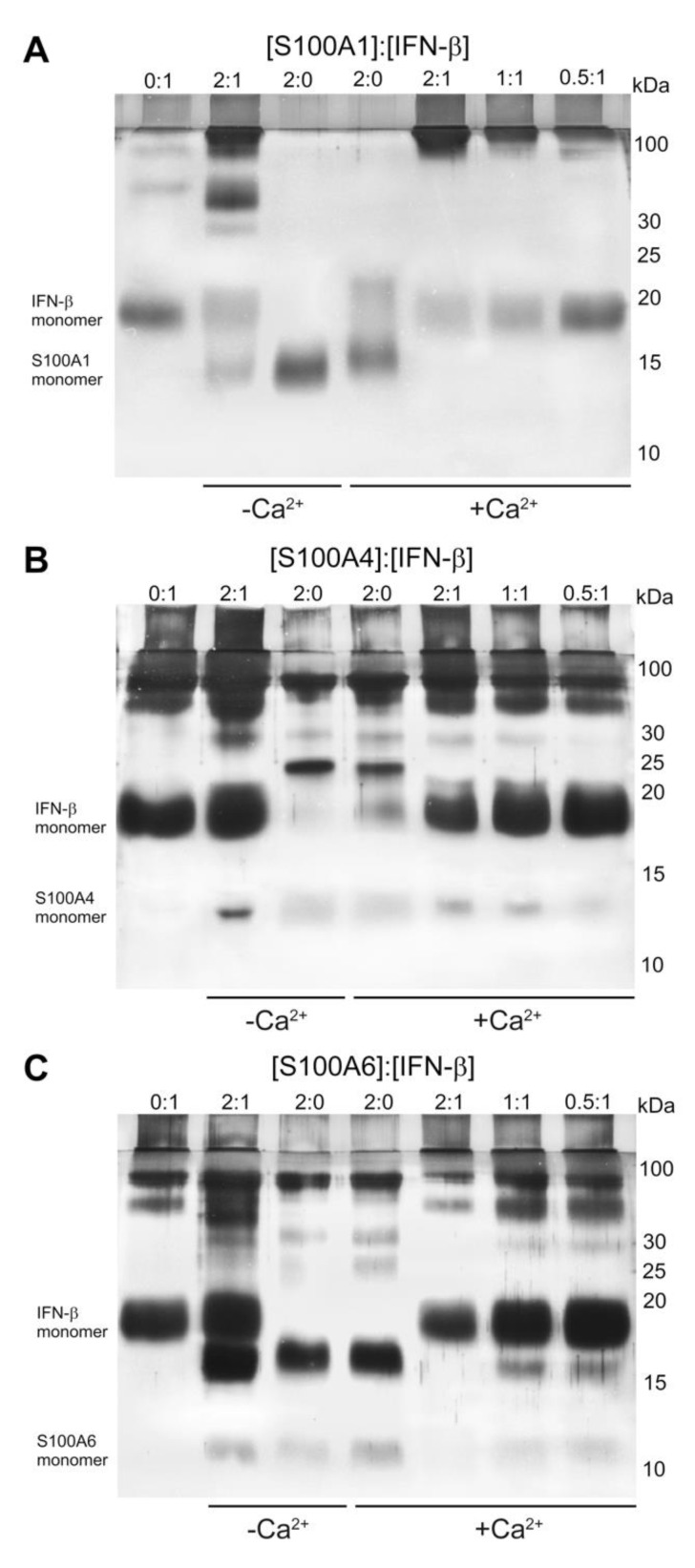
The results of SDS-PAGE for 5 μM IFN-β with Ca^2+^-free/bound S100A1 (panel **A**), S100A4 (**B**) or S100A6 (**C**) (S100 to IFN-β molar ratios are indicated), after crosslinking with EDAC/sulfo-NHS at 25 °C.

**Figure 4 ijms-21-09473-f004:**
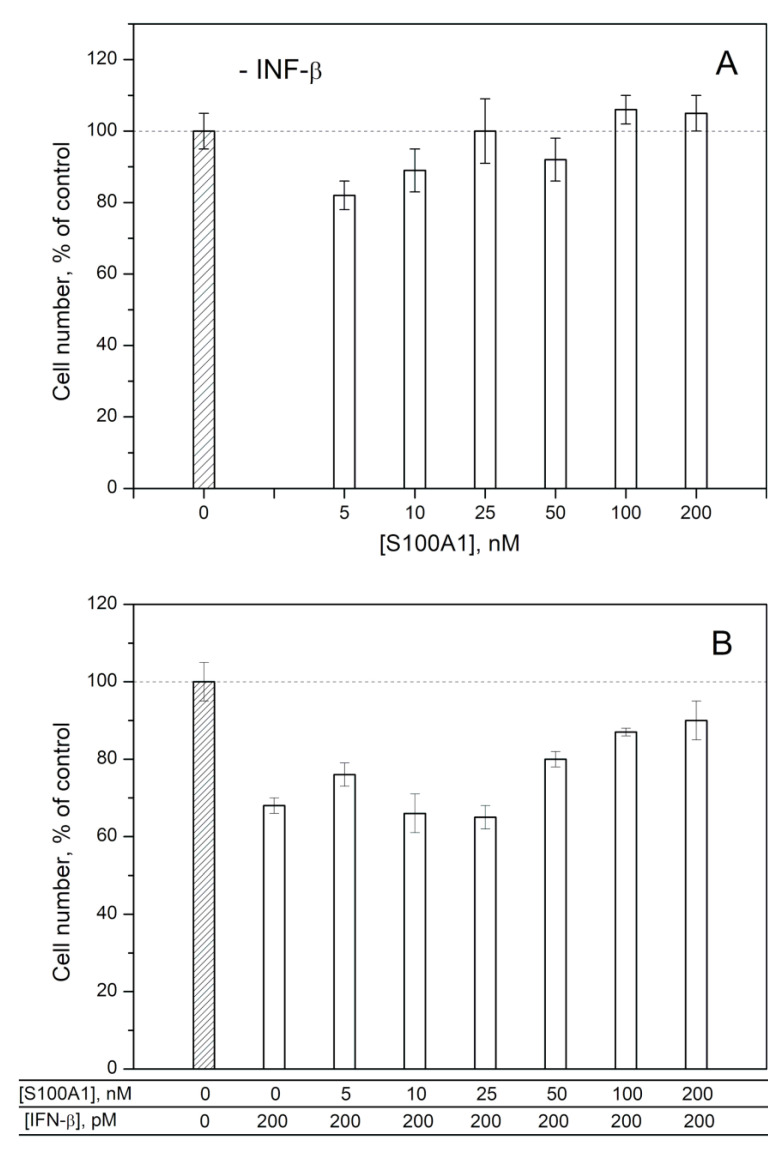
Viability of MCF-7 cells treated by S100A1 protein (panel **A**), IFN-β or their combination (**B**) measured by crystal violet assay.

**Figure 5 ijms-21-09473-f005:**
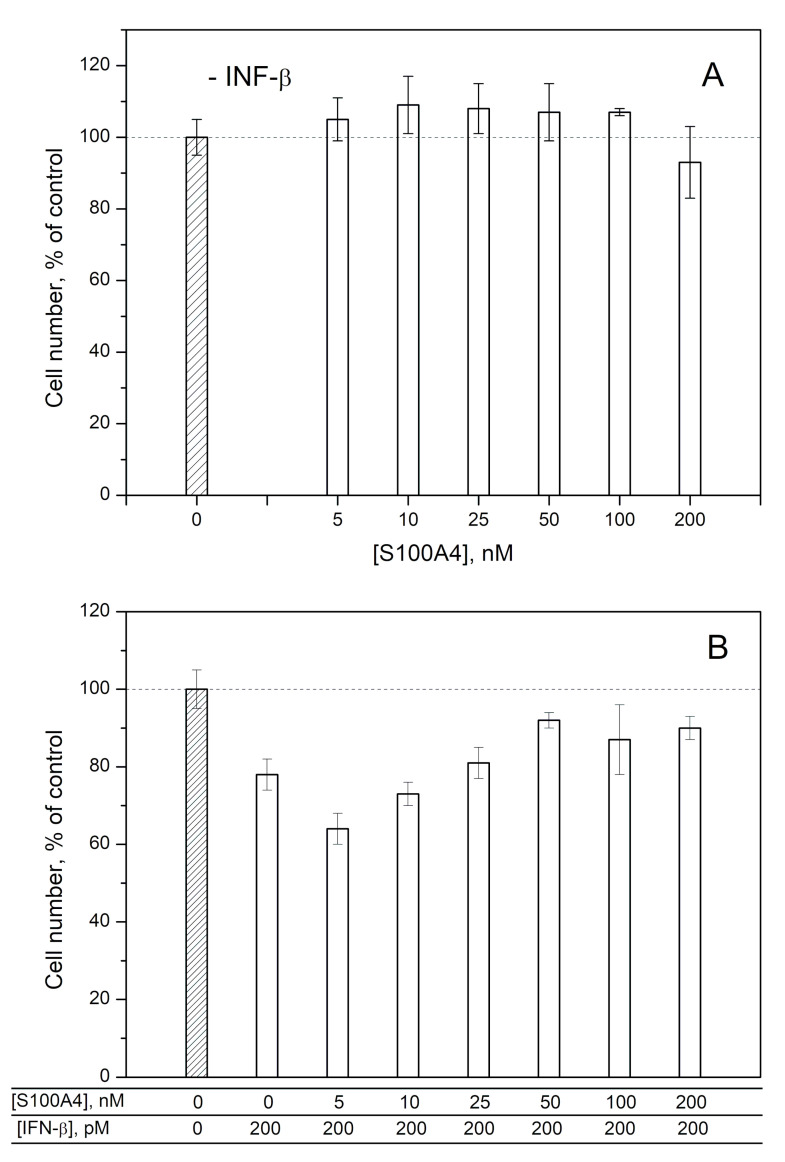
Viability of MCF-7 cells treated by S100A4 protein (panel **A**), IFN-β or their combination (**B**) measured by crystal violet assay.

**Figure 6 ijms-21-09473-f006:**
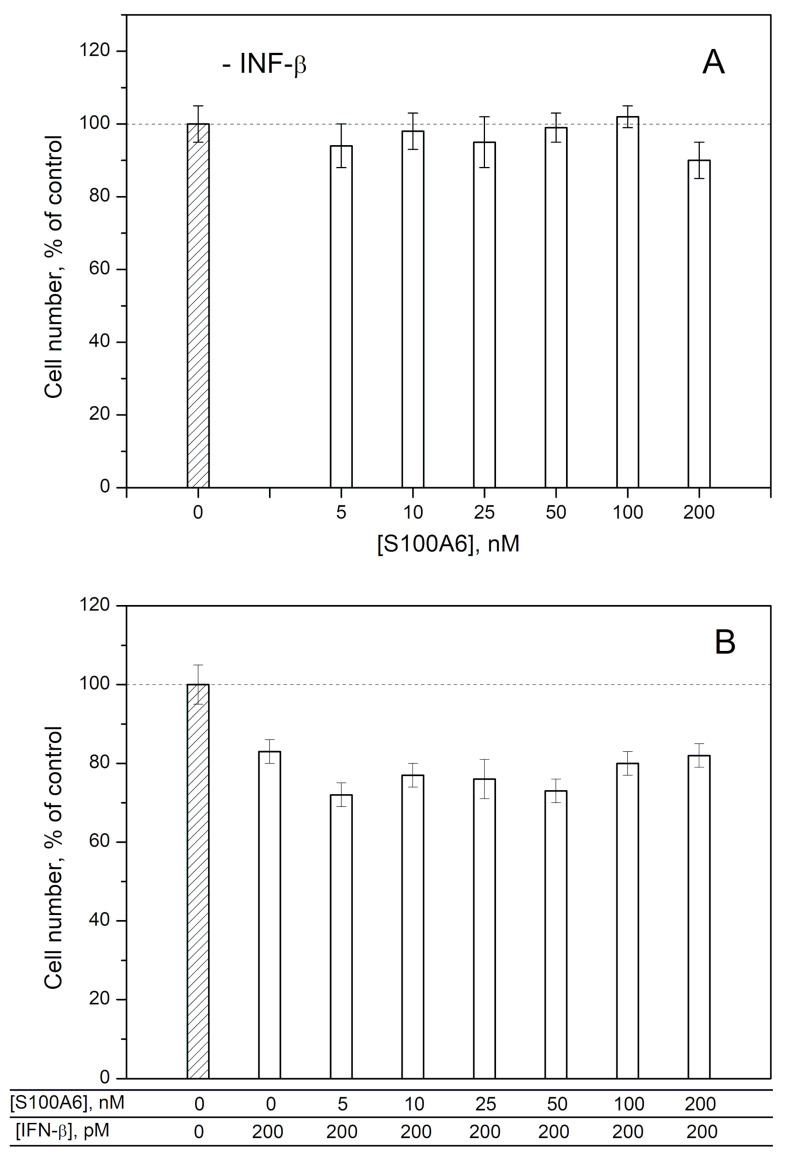
Viability of MCF-7 cells treated by S100A6 protein (panel **A**), IFN-β or their combination (**B**) measured by crystal violet assay.

**Figure 7 ijms-21-09473-f007:**
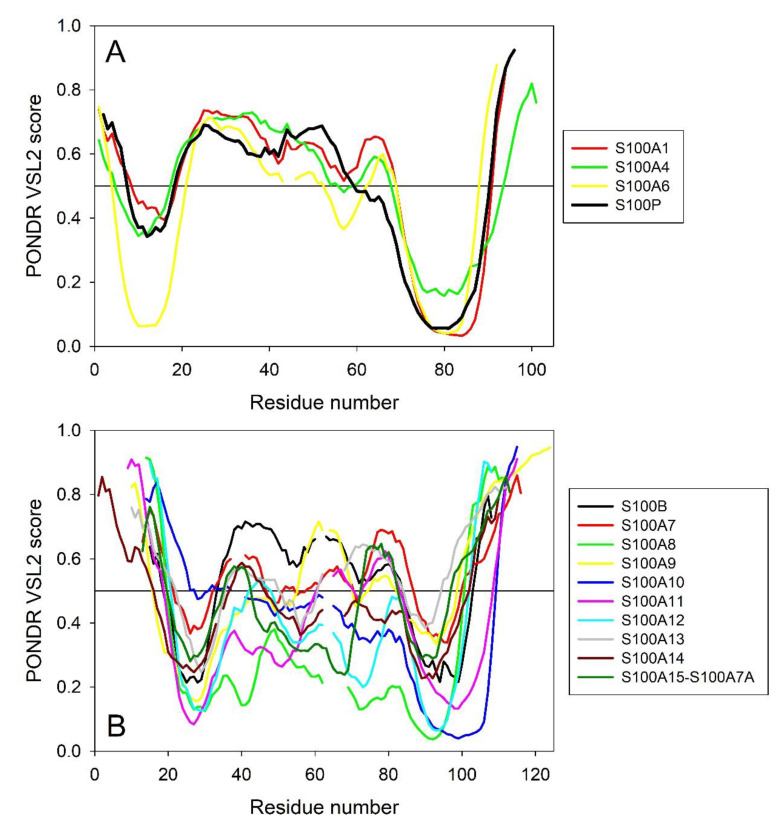
Intrinsic disorder predisposition of human S100 proteins capable of IFN-β binding (**A**) and non-binders (**B**) evaluated by the PONDR^®^ VSL2 algorithm. Plot represents data for the sequences aligned using Clustal Omega. Breaks in the lines corresponds to the alignment gaps.

**Figure 8 ijms-21-09473-f008:**
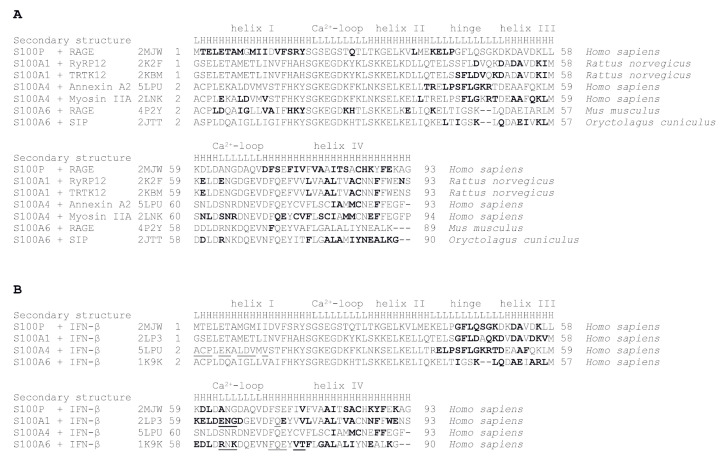
Amino acid sequence alignment for the S100 proteins specific to IFN-β, derived from overlay of their tertiary structures on that for human S100P using DALI server [59]. The residues of the primary monomers of the S100 proteins involved in the interaction with either target proteins (panel **A**) or human IFN-β (**B**), determined with LPC software [60], are shown in bold. Model structures of the S100—IFN-β complexes are shown in Figure 8. The residues of the secondary monomers of the S100 proteins predicted to be involved in the interaction with IFN-β are underlined (**B**). PDB codes used for the analysis and secondary structure elements (“L” and “H” denote loops and α-helices, respectively) are indicated.

**Figure 9 ijms-21-09473-f009:**
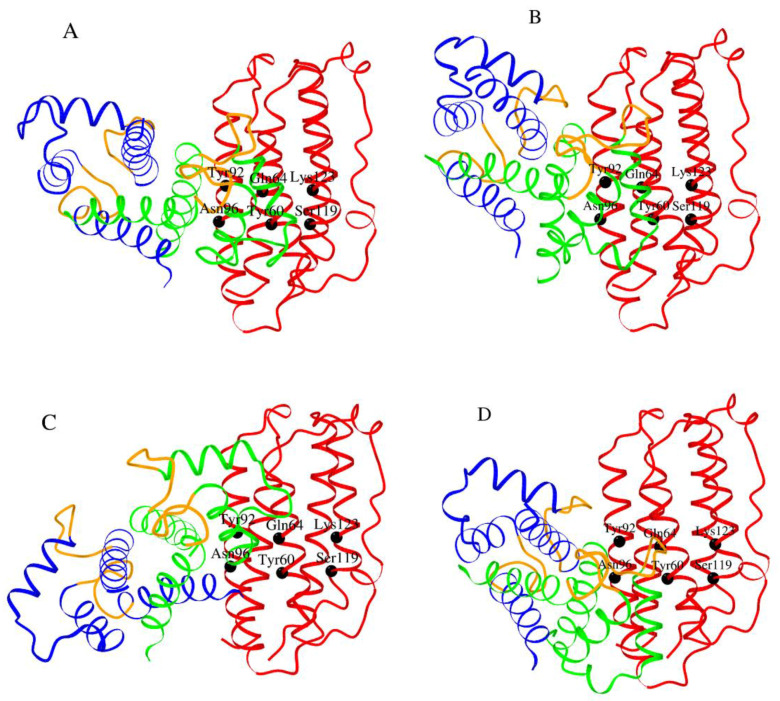
The models of tertiary structures of Ca^2+^-loaded human S100P (panel **A**), S100A1 (**B**), S100A4 (**C**) and S100A6 (**D**) dimers (the monomers are shown in blue/green) bound to a molecule of human IFN-β (red) built using GRAMM-X protein–protein docking software v.1.2.0 [58]. The Ca^2+^-binding loops are shown in orange. The contact residues of IFN-β are labelled.

**Table 1 ijms-21-09473-t001:** Parameters of the heterogeneous ligand model [1] describing the SPR data on kinetics of the IFN-β—S100 interactions (Figure 1 and Figure 2). *R_max_* refers to maximum response values. The standard deviations are indicated.

Analyte	Ligand	*k_d1_*, s^−1^	*K_d1_*, nM	*R_max1_*	*k_d2_*, s^−1^	*K_d2_*, nM	*R_max2_*
S100A1	IFN-β	(2.71 ± 0.13) × 10^−4^	41 ± 12	190	(3.65 ± 0.54) × 10^−3^	1450 ± 544	68
S100A4		(5.97 ± 0.22) × 10^−4^	227 ± 19	98	(1.08 ± 0.53) × 10^−2^	443 ± 249	38
S100A6		(2.10 ± 0.57) × 10^−4^	82 ± 24	313	(4.07 ± 0.97) × 10^−3^	267 ± 57	81
IFN-β	S100A1	(6.46 ± 0.10) × 10^−3^	47 ± 10	361	(6.80 ± 0.53) × 10^−5^	0.11 ± 0.06	108
	S100A4	(1.75 ± 0.24) × 10^−2^	105 ± 11	112	(8.03 ± 0.26) × 10^−5^	1.0 ± 0.1	19
	S100A6	(2.10 ± 0.57) × 10^−5^	0.70 ± 0.03	43	(2.60 ± 0.27) × 10^−2^	281 ± 49	20

**Table 2 ijms-21-09473-t002:** List of the human diseases associated with IFN-β and S100A1, S100A4, S100A6 or S100P protein, according to Open Targets Platform (https://www.opentargets.org). The association scores for IFN-β and S100 exceed 0.1.

Diseases Associated with S100A1 and IFN-β	Diseases Associated with S100A4 and IFN-β	Diseases Associated with S100A6 and IFN-β	Diseases Associated with S100P and IFN-β
neoplasm, cancer, carcinoma (lung, adenocarcinoma), infectious disease, vascular disease, pulmonary arterial hypertension, nervous system disease, heart disease, cardiomyopathy, hypertension	neoplasm (liver, lung, breast, skin, brain, ovarian), glioma, astrocytoma, cancer (lung, central nervous system, breast), carcinoma (non-small cell lung, lung, ovarian, breast, adenocarcinoma), glioblastoma multiforme, infectious disease, immune system disease, vascular disease, nervous system disease, neuropathy, lung disease, liver disease	neoplasm (lung), cancer (lung), carcinoma (lung), nervous system disease, lung disease	neoplasm (liver, brain), cancer (lung), carcinoma, adenocarcinoma, glioblastoma multiforme, respiratory system disease, liver disease

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
