# Peer review of "Interferon Beta Activity Is Modulated via Binding of Specific S100 Proteins"

_ijms, 2020, doi:10.3390/ijms21249473_

Round 1

Reviewer 1 Report

This manuscript describes the interaction between S100 protein and interferon beta, and the regulation of interferon beta activity by S100 protein.

The reviewer is concerned about the following:

1) The affinity between S100 and interferon beta indicates a discrepancy between the analyte and ligand pair. Table 1 shows the two affinity constants for each pair, based on a heterogeneous ligand model that assumes two populations of ligands (L1 and L2). What are the chemical entities of L1 and L2 (for Kd1 and Kd2) in Table 1?

2) The in vivo activity of the S100 protein shown in Fig. 4 does not correspond to the binding affinity shown in Table 1. The authors suggested that a three-component complex of IFNAR1/IFNAR2 and interferon may inhibit the interaction between S100 and interferon. In experiments with the S100 and interferon premix, the effects of such steric hindrance will be small.

3) In vivo inhibition by S100 is weak. Normally, the concentration of S100 protein in the blood is about 0.1-10 nM even under elevated conditions. There is no case where the viability of MCF-7 is restored even with S100 of 200nM.

Author Response

This manuscript describes the interaction between S100 protein and interferon beta, and the regulation of interferon beta activity by S100 protein.

The reviewer is concerned about the following:

1) The affinity between S100 and interferon beta indicates a discrepancy between the analyte and ligand pair. Table 1 shows the two affinity constants for each pair, based on a heterogeneous ligand model that assumes two populations of ligands (L1 and L2). What are the chemical entities of L1 and L2 (for Kd1 and Kd2) in Table 1?

RESPONSE: The ligand is chemically linked to the surface of SPR chip via amino groups, which suggests the possibility that the ligand can be exposed to the analyte in several preferential conformations, differing by their ability to recognize the analyte. Some of the conformations may preclude the interaction with the analyte due to sterical hindrance, while others are characterized by “open” conformation. L1 and L2 correspond to “open” conformations with different affinity to the analyte. They could correspond to different analyte-binding sites or different conformations of the same site. Although this is very typical for SPR experiments, it is rarely possible to distinguish these situations. The corresponding clarification is added to the revised manuscript (see lines 479-487).

2) The in vivo activity of the S100 protein shown in Fig. 4 does not correspond to the binding affinity shown in Table 1. The authors suggested that a three-component complex of IFNAR1/IFNAR2 and interferon may inhibit the interaction between S100 and interferon. In experiments with the S100 and interferon premix, the effects of such steric hindrance will be small.

RESPONSE: We see no any evident discrepancies between our estimates of IFN-b affinities to the S100 proteins and the results of the cellular assays. In fact, we see their correlation: the half-transitions for viability of MCF-7 cells are observed at S100A1/A4 concentrations of 50 nM, which corresponds to the S100A4-IFN-b equilibrium dissociation constants (Kd) of 41/47 nM, and in-between the estimates of Kd values for IFN-b interaction with monomeric S100A4 (1/105 nM). The corresponding clarification is added to the revised manuscrtipt (lines 202-207).

3) In vivo inhibition by S100 is weak. Normally, the concentration of S100 protein in the blood is about 0.1-10 nM even under elevated conditions. There is no case where the viability of MCF-7 is restored even with S100 of 200nM.

RESPONSE: Indeed, the suppression is incomplete. To emphasize this fact, the phrase (lines 196-197) ”Hence, S100A1-bound IFN-β has suppressed cytotoxicity against the MCF-7 cells” has been corrected: “Hence, S100A1-bound IFN-β has partially suppressed cytotoxicity against the MCF-7 cells”.

Reviewer 2 Report

The authors describe the interactions between interferon-beta and three calcium ion dependent S100 proteins where their binding inhibits anticancer activity of interferon-beta. The authors present clear evidence of binding that inhibits the suppression of MCF-7 cell line viability. They also show a regulatory role of S100 proteins. The evidence is convincing and the their methods are sound. This reviewer recommends publication.

Author Response

The authors describe the interactions between interferon-beta and three calcium ion dependent S100 proteins where their binding inhibits anticancer activity of interferon-beta. The authors present clear evidence of binding that inhibits the suppression of MCF-7 cell line viability. They also show a regulatory role of S100 proteins. The evidence is convincing and their methods are sound. This reviewer recommends publication.

RESPONSE: We are thankful to this reviewer for high evaluation of our work.

Round 2

Reviewer 1 Report

The reviewer added the COMMENTS.

This manuscript describes the interaction between S100 protein and interferon beta, and the regulation of interferon beta activity by S100 protein.

The reviewer is concerned about the following:

1) The affinity between S100 and interferon beta indicates a discrepancy between the analyte and ligand pair. Table 1 shows the two affinity constants for each pair, based on a heterogeneous ligand model that assumes two populations of ligands (L1 and L2). What are the chemical entities of L1 and L2 (for Kd1 and Kd2) in Table 1?

RESPONSE: The ligand is chemically linked to the surface of SPR chip via amino groups, which suggests the possibility that the ligand can be exposed to the analyte in several preferential conformations, differing by their ability to recognize the analyte. Some of the conformations may preclude the interaction with the analyte due to sterical hindrance, while others are characterized by “open” conformation. L1 and L2 correspond to “open” conformations with different affinity to the analyte. They could correspond to different analyte-binding sites or different conformations of the same site. Although this is very typical for SPR experiments, it is rarely possible to distinguish these situations. The corresponding clarification is added to the revised manuscript (see lines 479-487).

COMMENTS:

Table 1.

S100 dimer and IFN

S100A1: 41-1450 nM

S100A4: 227-443 nM

S100A6: 82-267 nM

S100 monomer and IFN

S100A1: 0.11-47 nM

S100A4: 1.0-105 nM

S100A6: 0.7-281 nM

How can we estimate the binding strength between each S100 and IFN in solution or in vivo from the values of Table 1?  Which is the strongest, which is the weakest?

2) The in vivo activity of the S100 protein shown in Fig. 4 does not correspond to the binding affinity shown in Table 1. The authors suggested that a three-component complex of IFNAR1/IFNAR2 and interferon may inhibit the interaction between S100 and interferon. In experiments with the S100 and interferon premix, the effects of such steric hindrance will be small.

RESPONSE: We see no any evident discrepancies between our estimates of IFN-b affinities to the S100 proteins and the results of the cellular assays. In fact, we see their correlation: the half-transitions for viability of MCF-7 cells are observed at S100A1/A4 concentrations of 50 nM, which corresponds to the S100A4-IFN-b equilibrium dissociation constants (Kd) of 41/47 nM, and in-between the estimates of Kd values for IFN-b interaction with monomeric S100A4 (1/105 nM). The corresponding clarification is added to the revised manuscrtipt (lines 202-207).

COMMENTS:

The binding affinities between each S100 and IFN are comparable.  S100A6 can interact to IFN as strong as S100A1 and S100A4 do.  Authors should discuss why S100A6 has no effects on the viability test with IFN against MCF-7 cells.

3) In vivo inhibition by S100 is weak. Normally, the concentration of S100 protein in the blood is about 0.1-10 nM even under elevated conditions. There is no case where the viability of MCF-7 is restored even with S100 of 200nM.

RESPONSE: Indeed, the suppression is incomplete. To emphasize this fact, the phrase (lines 196-197) ”Hence, S100A1-bound IFN-β has suppressed cytotoxicity against the MCF-7 cells” has been corrected: “Hence, S100A1-bound IFN-β has partially suppressed cytotoxicity against the MCF-7 cells”.

COMMENTS:

It is necessary to discuss whether the inhibitory activity of S100 protein on IFN cytotoxicity has physiological significance despite this degree of strength.

Author Response

This manuscript describes the interaction between S100 protein and interferon beta, and the regulation of interferon beta activity by S100 protein.

The reviewer is concerned about the following:

1) The affinity between S100 and interferon beta indicates a discrepancy between the analyte and ligand pair. Table 1 shows the two affinity constants for each pair, based on a heterogeneous ligand model that assumes two populations of ligands (L1 and L2). What are the chemical entities of L1 and L2 (for Kd1 and Kd2) in Table 1?

RESPONSE: The ligand is chemically linked to the surface of SPR chip via amino groups, which suggests the possibility that the ligand can be exposed to the analyte in several preferential conformations, differing by their ability to recognize the analyte. Some of the conformations may preclude the interaction with the analyte due to sterical hindrance, while others are characterized by “open” conformation. L1 and L2 correspond to “open” conformations with different affinity to the analyte. They could correspond to different analyte-binding sites or different conformations of the same site. Although this is very typical for SPR experiments, it is rarely possible to distinguish these situations. The corresponding clarification is added to the revised manuscript (see lines 479-487).

COMMENTS:

Table 1.

S100 dimer and IFN

S100A1: 41-1450 nM

S100A4: 227-443 nM

S100A6: 82-267 nM

S100 monomer and IFN

S100A1: 0.11-47 nM

S100A4: 1.0-105 nM

S100A6: 0.7-281 nM

How can we estimate the binding strength between each S100 and IFN in solution or in vivo from the values of Table 1?  Which is the strongest, which is the weakest?

RESPONSE: The Kd values shown in Table 1 correspond to in vitro interaction between IFN-b and S100 proteins: the lower the Kd values, the stronger the interaction.

2) The in vivo activity of the S100 protein shown in Fig. 4 does not correspond to the binding affinity shown in Table 1. The authors suggested that a three-component complex of IFNAR1/IFNAR2 and interferon may inhibit the interaction between S100 and interferon. In experiments with the S100 and interferon premix, the effects of such steric hindrance will be small.

RESPONSE: We see no any evident discrepancies between our estimates of IFN-b affinities to the S100 proteins and the results of the cellular assays. In fact, we see their correlation: the half-transitions for viability of MCF-7 cells are observed at S100A1/A4 concentrations of 50 nM, which corresponds to the S100A4-IFN-b equilibrium dissociation constants (Kd) of 41/47 nM, and in-between the estimates of Kd values for IFN-b interaction with monomeric S100A4 (1/105 nM). The corresponding clarification is added to the revised manuscrtipt (lines 202-207).

COMMENTS:

The binding affinities between each S100 and IFN are comparable.  S100A6 can interact to IFN as strong as S100A1 and S100A4 do.  Authors should discuss why S100A6 has no effects on the viability test with IFN against MCF-7 cells.

RESPONSE: We have inserted the following considerations into the text (lines 207-209): “Meanwhile, exogenous S100A6 does not induce well-defined changes in viability of MCF-7 cells (Figure 6), which indicates that structural peculiarities of IFN-β - S100A6 complex differ from those for IFN-β - S100A1/A4 complexes”.

Furthermore, the lines 326-329 provide a more mechanistic explanation of the results of the cellular assays: “the differences in relative orientations of S100A1/A4/A6/P molecules within the models of complexes with IFN-β could explain their different abilities to inhibit the IFN-β-induced suppression of MCF-7 cells viability”.

3) In vivo inhibition by S100 is weak. Normally, the concentration of S100 protein in the blood is about 0.1-10 nM even under elevated conditions. There is no case where the viability of MCF-7 is restored even with S100 of 200nM.

RESPONSE: Indeed, the suppression is incomplete. To emphasize this fact, the phrase (lines 196-197) ”Hence, S100A1-bound IFN-β has suppressed cytotoxicity against the MCF-7 cells” has been corrected: “Hence, S100A1-bound IFN-β has partially suppressed cytotoxicity against the MCF-7 cells”.

COMMENTS:

It is necessary to discuss whether the inhibitory activity of S100 protein on IFN cytotoxicity has physiological significance despite this degree of strength.

RESPONSE: We have added the following discussion into the text (lines 215-219):

“The half-transition S100A1/A4 concentration in the MCF-7 viability assays (Figures 4-5) of 50 nM exceeds the serum S100A1/A4 levels under pathological conditions of 5-11 nM (may reach 95 nM for S100A4) [48, 49]. Meanwhile, local S100A1/A4 concentrations within sites of the pathological processes are expected to be substantially higher, thereby favoring development of the cellular effects observed under the in vitro conditions”.